# Defense Against Adversarial Attacks Using Feature Scattering-based Adversarial Training

**Haichao Zhang**[*]    **Jianyu Wang**
Horizon Robotics    Baidu Research
hczhang1@gmail.com    wjyouch@gmail.com

## Abstract

We introduce a feature scattering-based adversarial training approach for improving model robustness against adversarial attacks. Conventional adversarial training approaches leverage a supervised scheme (either targeted or non-targeted) in generating attacks for training, which typically suffer from issues such as label leaking as noted in recent works. Differently, the proposed approach generates adversarial images for training through feature scattering in the latent space, which is unsupervised in nature and avoids label leaking. More importantly, this new approach generates perturbed images in a collaborative fashion, taking the inter-sample relationships into consideration. We conduct analysis on model robustness and demonstrate the effectiveness of the proposed approach through extensively experiments on different datasets compared with state-of-the-art approaches. Code is available: https://github.com/Haichao-Zhang/FeatureScatter.

## 1 Introduction

While breakthroughs have been made in many fields such as image classification leveraging deep neural networks, these models could be easily fooled by the so call adversarial examples [55, 4]. In terms of the image classification, an adversarial example for a natural image is a modified version which is visually indistinguishable from the original but causes the classifier to produce a different label prediction [4, 55, 24]. Adversarial examples have been shown to be ubiquitous beyond classification, ranging from object detection [64, 18] to speech recognition [11, 9].

Many encouraging progresses been made towards improving model robustness against adversarial examples under different scenarios [58, 36, 33, 67, 72, 16, 71]. Among them, adversarial training [24, 36] is one of the most popular technique [2], which conducts model training using the adversarially perturbed images in place of the original ones. However, several challenges remain to be addressed. Firstly, some adverse effects such as label leaking is still an issue hindering adversarial training [32]. Currently available remedies either increase the number of iterations for generating the attacks [36] or use classes other than the ground-truth for attack generation [32, 65, 61]. Increasing the attack iterations will increase the training time proportionally while using non-ground-truth targeted approach cannot fully eliminate label leaking. Secondly, previous approaches for both standard and adversarial training treat each training sample individually and in isolation *w.r.t.* other samples. Manipulating each sample individually this way neglects the inter-sample relationships and does not fully leverage the potential for attacking and defending, thus limiting the performance.

Manifold and neighborhood structure have been proven to be effective in capturing the inter-sample relationships [51, 22]. Natural images live on a low-dimensional manifold, with the training and testing images as samples from it [26, 51, 44, 56]. Modern classifiers are over-complete in terms of parameterizations and different local minima have been shown to be equally effective under the clean image setting [14]. However, different solution points might leverage different set of features for

---

[*]Work done while with Baidu Research.

prediction. For learning a well-performing classifier on natural images, it suffices to simply adjust the classification boundary to intersect with this manifold at locations with good separation between classes on training data, as the test data will largely reside on the same manifold [28]. However, the classification boundary that extends beyond the manifold is less constrained, contributing to the existence of adversarial examples [56, 59]. For examples, it has been pointed out that some clean trained models focus on some discriminative but less robust features, thus are vulnerable to adversarial attacks [28, 29]. Therefore, the conventional supervised attack that tries to move feature points towards this decision boundary is likely to disregard the original data manifold structure. When the decision boundary lies close to the manifold for its out of manifold part, adversarial perturbations lead to a *tilting* effect on the data manifold [56]; at places where the classification boundary is far from the manifold for its out of manifold part, the adversarial perturbations will move the points towards the decision boundary, effectively *shrinking* the data manifold. As the adversarial examples reside in a large, contiguous region and a significant portion of the adversarial subspaces is shared [24, 19, 59, 40], pure label-guided adversarial examples will *clutter* as least in the shared adversarial subspace. In summary, while these effects encourage the model to focus more around the current decision boundary, they also make the effective data manifold for training deviate from the original one, potentially hindering the performance.

Motived by these observations, we propose to shift the previous focus on the *decision boundary* to the *inter-sample structure*. The proposed approach can be intuitively understood as generating adversarial examples by perturbing the local neighborhood structure in an *unsupervised* fashion and then performing model training with the generated adversarial images. The overall framework is shown in Figure 1. The contributions of this work are summarized as follows:

- we propose a novel feature-scattering approach for generating adversarial images for adversarial training in a collaborative and unsupervised fashion;
- we present an adversarial training formulation which deviates from the conventional minimax formulation and falls into a broader category of *bilevel optimization*;
- we analyze the proposed approach and compare it with several state-of-the-art techniques, with extensive experiments on a number of standard benchmarks, verifying its effectiveness.

## 2 Background

### 2.1 Adversarial Attack, Defense and Adversarial Training

Adversarial examples, initially demonstrated in [4, 55], have attracted great attention recently [4, 24, 58, 36, 2, 5]. Szegedy *et al.* pointed out that CNNs are vulnerable to adversarial examples and proposed an L-BFGS-based algorithm for generating them [55]. A fast gradient sign method (FGSM) for adversarial attack generation is developed and used in adversarial training in [24]. Many variants of attacks have been developed later [41, 8, 54, 62, 7, 6]. In the mean time, many efforts have been devoted to defending against adversarial examples [38, 37, 63, 25, 33, 50, 53, 46, 35]. Recently, [2] showed that many existing defence methods suffer from a false sense of robustness against adversarial attacks due to gradient masking, and adversarial training [24, 32, 58, 36] is one of the effective defense method against adversarial attacks. It improves model robustness by solving a minimax problem as [24, 36]:

$$\min_{\boldsymbol{\theta}} \left[ \max_{\mathbf{x}' \in \mathcal{S}_{\mathbf{x}}} \mathcal{L}(\mathbf{x}', y; \boldsymbol{\theta}) \right] \tag{1}$$

where the inner maximization essentially generates attacks while the outer minimization corresponds to minimizing the "adversarial loss" induced by the inner attacks [36]. The inner maximization can be solved approximately, using for example a one-step approach such as FGSM [24], or a multi-step projected gradient descent (PGD) method [36]

$$\mathbf{x}^{t+1} = \mathcal{P}_{\mathcal{S}_{\mathbf{x}}}\left(\mathbf{x}^t + \alpha \cdot \text{sign}\left(\nabla_{\mathbf{x}} \mathcal{L}(\mathbf{x}^t, y; \boldsymbol{\theta})\right)\right), \tag{2}$$

where $\mathcal{P}_{\mathcal{S}_{\mathbf{x}}}(\cdot)$ is a projection operator projecting the input into the feasible region $\mathcal{S}_{\mathbf{x}}$. In the PGD approach, the original image $\mathbf{x}$ is randomly perturbed to some point $\mathbf{x}^0$ within $B(\mathbf{x}, \epsilon)$, the $\epsilon$-cube around $\mathbf{x}$, and then goes through several PGD steps with a step size of $\alpha$ as shown in Eqn.(2).

Label leaking [32] and gradient masking [43, 58, 2] are some well-known issues that hinder the adversarial training [32]. Label leaking occurs when the additive perturbation is highly correlated with the ground-truth label. Therefore, when it is added to the image, the network can directly tell the class label by decoding the additive perturbation without relying on the real content of the image, leading

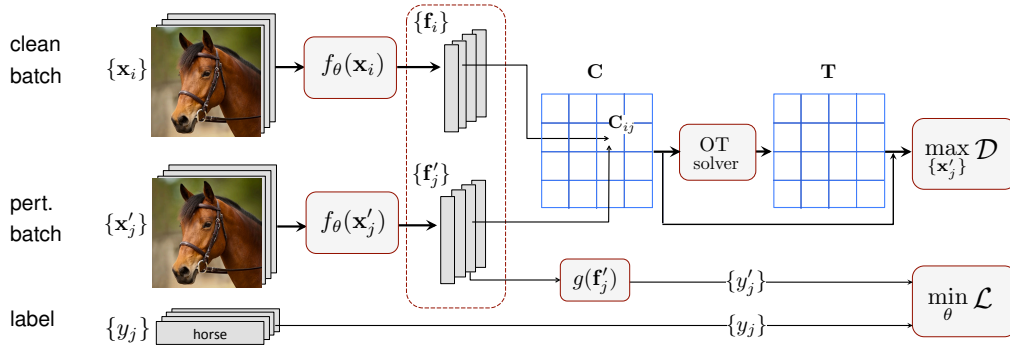

Figure 1: **Feature Scattering-based Adversarial Training Pipeline**. The adversarial perturbations are generated collectively by feature scattering, *i.e.*, maximizing the feature matching distance between the clean samples $\{\mathbf{x}_i\}$ and the perturbed samples $\{\mathbf{x}'_j\}$. The model parameters are updated by minimizing the cross-entropy loss using the perturbed images $\{\mathbf{x}'_j\}$ as the training samples.

to higher adversarial accuracy than the clean image during training. Gradient masking [43, 58, 2] refers to the effect that the adversarially trained model learns to "improve" robustness by generating less useful gradients for adversarial attacks, which could be by-passed with a substitute model for generating attacks, thus giving a false sense of robustness [2].

## 2.2 Different Distances for Feature and Distribution Matching

Euclidean distance is arguably one of the most commonly used metric for measuring the distance between a pair of points. When it comes to two sets of points, it is natural to accumulate the individual pairwise distance as a measure of distance between the two sets, given the proper correspondence. Alternatively, we can view each set as an empirical distribution and measure the distance between them using *Kullback-Leibler* (KL) or Jensen-Shannon (JS) divergence. The challenge for learning with KL or JS divergence is that no useful gradient is provided when the two empirical distributions have disjoint supports or have a non-empty intersection contained in a set of measure zero [1, 49]. The optimal transport (OT) distance is an alternative measure of the distance between distributions with advantages over KL and JS in the scenarios mentioned earlier. The OT distance between two probability measures $\mu$ and $\nu$ is defined as:

$$\mathcal{D}(\mu, \nu) = \inf_{\gamma \in \Pi(\mu,\nu)} \mathbb{E}_{(x,y)\sim\gamma} \; c(x, y) \,, \tag{3}$$

where $\Pi(\mu, \nu)$ denotes the set of all joint distributions $\gamma(x, y)$ with marginals $\mu(x)$ and $\nu(y)$, and $c(x, y)$ is the cost function (Euclidean or cosine distance). Intuitively, $\mathcal{D}(\mu, \nu)$ is the minimum cost that $\gamma$ has to transport from $\mu$ to $\nu$. It provides a weaker topology than many other measures, which is important for applications where the data typically resides on a low dimensional manifold of the input embedding space [1, 49], which is the case for natural images. It has been widely applied to many tasks, such as generative modeling [21, 1, 49, 20, 10], auto-encoding [57] and dictionary learning [47]. For comprehensive historical and computational perspective of OT, we refer to [60, 45].

# 3 Feature Scattering-based Adversarial Training

## 3.1 Feature Matching and Feature Scattering

**Feature Matching**. Conventional training treats training data as *i.i.d* samples from a data distribution, overlooking the connections between samples. The same assumption is used when generating adversarial examples for training, with the direction for perturbing a sample purely based on the direction from the current data point to the decision boundary, regardless of other samples. While effective, it disregards the inter-relationship between different feature points, as the adversarial perturbation is computed individually for each sample, neglecting any collective distributional property. Furthermore, the supervised generation of the attacks makes the generated perturbations highly biases towards the decision boundary, as shown in Figure 2. This is less desirable as it might neglect other directions that are crucial for learning robust models [28, 17] and leads to label leaking due to high correlation between the perturbation and the decision boundary.

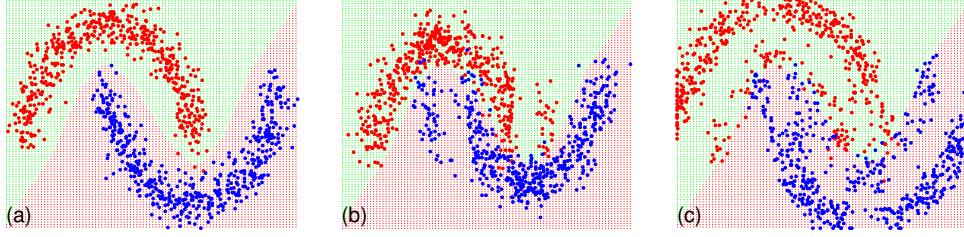

Figure 2: **Illustration Example of Different Perturbation Schemes**. (a) Original data. Perturbed data using (b) *supervised* adversarial generation method and (c) the proposed *feature scattering*, which is an *unsupervised* method. The overlaid boundary is from the model trained on clean data.

The idea of leveraging inter-sample relationship for learning dates back to the seminal work of [51, 22, 48]. This type of local structure is also exploited in this work, but for adversarial perturbation. The quest of local structure utilization and seamless integration with the end-to-end-training framework naturally motivates an OT-based soft matching scheme, using the OT-distance as in Eqn.(3). We consider OT between discrete distributions hereafter as we mainly focus on applying the OT distance on image features. Specifically, consider two discrete distributions $\boldsymbol{\mu}, \boldsymbol{\nu} \in \mathbb{P}(\mathbb{X})$, which can be written as $\boldsymbol{\mu} = \sum_{i=1}^{n} u_i \delta_{\mathbf{x}_i}$ and $\boldsymbol{\nu} = \sum_{i=1}^{n} v_i \delta_{\mathbf{x}'_i}$, with $\delta_{\mathbf{x}}$ the Dirac function centered on $\mathbf{x}$.[2] The weight vectors $\boldsymbol{\mu} = \{u_i\}_{i=1}^{n} \in \Delta_n$ and $\boldsymbol{\nu} = \{v_i\}_{i=1}^{n} \in \Delta_n$ belong to the $n$-dimensional simplex, *i.e.*, $\sum_i u_i = \sum_i v_i = 1$, as both $\boldsymbol{\mu}$ and $\boldsymbol{\nu}$ are probability distributions. Under such a setting, computing the OT distance as defined in Eqn.(3) is equivalent to solving the following network-flow problem

$$\mathcal{D}(\boldsymbol{\mu}, \boldsymbol{\nu}) = \min_{\mathbf{T} \in \Pi(\mathbf{u}, \mathbf{v})} \sum_{i=1}^{n} \sum_{j=1}^{n} \mathbf{T}_{ij} \cdot c(\mathbf{x}_i, \mathbf{x}'_j) = \min_{\mathbf{T} \in \Pi(\mathbf{u}, \mathbf{v})} \langle \mathbf{T}, \mathbf{C} \rangle \qquad (4)$$

where $\Pi(\mathbf{u}, \mathbf{v}) = \{\mathbf{T} \in \mathbb{R}_+^{n \times n} | \mathbf{T} \mathbf{1}_n = \mathbf{u}, \mathbf{T}^\top \mathbf{1}_n = \mathbf{v}\}$. $\mathbf{1}_n$ is an $n$-dimensional all-one vector. $\langle \cdot, \cdot \rangle$ represents the Frobenius dot-product. $\mathbf{C}$ is the transport cost matrix such that $\mathbf{C}_{ij} = c(\mathbf{x}_i, \mathbf{x}'_j)$. In this work, the transport cost is defined as the cosine distance between image features:

$$c(\mathbf{x}_i, \mathbf{x}'_j) = 1 - \frac{f_\theta(\mathbf{x}_i)^\top f_\theta(\mathbf{x}'_j)}{\|f_\theta(\mathbf{x}_i)\|_2 \|f_\theta(\mathbf{x}'_j)\|_2} = 1 - \frac{\mathbf{f}_i^\top \mathbf{f}'_j}{\|\mathbf{f}_i\|_2 \|\mathbf{f}'_j\|_2} \qquad (5)$$

where $f_\theta(\cdot)$ denotes the feature extractor with parameter $\theta$. We implement $f_\theta(\cdot)$ as the deep neural network upto the softmax layer. We can now formally define the feature matching distance as follows.

**Definition 1.** (Feature Matching Distance) *The feature matching distance between two set of images is defined as $\mathcal{D}(\boldsymbol{\mu}, \boldsymbol{\nu})$, the OT distance between empirical distributions $\boldsymbol{\mu}$ and $\boldsymbol{\nu}$ for the two sets.*

Note that the feature-matching distance is also a function of $\boldsymbol{\theta}$ (*i.e.* $\mathcal{D}_{\boldsymbol{\theta}}$) when $f_\theta(\cdot)$ is used for extracting the features in the computation of the ground distance as in Eqn.(5). We will simply use the notation $\mathcal{D}$ in the following when there is no danger of confusion to minimize notional clutter .

**Feature Scattering**. Based on the feature matching distance defined above, we can formulate proposed feature scattering method as follows:

$$\hat{\boldsymbol{\nu}} = \arg \max_{\boldsymbol{\nu} \in \mathcal{S}_{\boldsymbol{\mu}}} \mathcal{D}(\boldsymbol{\mu}, \boldsymbol{\nu}), \quad \boldsymbol{\mu} = \sum_{i=1}^{n} u_i \delta_{\mathbf{x}_i}, \ \boldsymbol{\nu} = \sum_{i=1}^{n} v_i \delta_{\mathbf{x}'_i}. \qquad (6)$$

This can be intuitively interpreted as maximizing the feature matching distance between the original and perturbed empirical distributions with respect to the inputs subject to domain constraints $\mathcal{S}_{\boldsymbol{\mu}}$

$$\mathcal{S}_{\boldsymbol{\mu}} = \{\sum_i v_i \delta_{\mathbf{z}_i}, | \ \mathbf{z}_i \in B(\mathbf{x}_i, \epsilon) \cap [0, 255]^d\},$$

where $B(\mathbf{x}, \epsilon) = \{\mathbf{z} \,|\, \|\mathbf{z} - \mathbf{x}\|_\infty \le \epsilon\}$ denotes the $\ell_\infty$-cube with center $\mathbf{x}$ and radius $\epsilon$. Formally, we present the notion of feature scattering as follows.

**Definition 2.** (Feature Scattering) *Given a set of clean data $\{\mathbf{x}_i\}$, which can be represented as an empirical distribution as $\boldsymbol{\mu} = \sum_i u_i \delta_{\mathbf{x}_i}$ with $\sum_i u_i = 1$, the feature scattering procedure is defined as producing a perturbed empirical distribution $\boldsymbol{\nu} = \sum_i v_i \delta_{\mathbf{x}'_i}$ with $\sum_i v_i = 1$ by maximizing $\mathcal{D}(\boldsymbol{\mu}, \boldsymbol{\nu})$, the feature matching distance between $\boldsymbol{\mu}$ and $\boldsymbol{\nu}$, subject to domain and budget constraints.*

**Remark 1.** *As the feature scattering is performed on a batch of samples leveraging inter-sample structure, it is more effective as adversarial attacks compared to structure-agnostic random perturbation while is less constrained than supervisedly generated perturbation which is decision boundary oriented and suffers from label leaking. Empirical comparisons will be provided in Section 5.*

## 3.2 Adversarial Training with Feature Scattering

We leverage feature scattering for adversarial training, with the mathmatical formulation as follows

$$\min_{\boldsymbol{\theta}} \frac{1}{n} \sum_{i=1}^{n} \mathcal{L}_{\boldsymbol{\theta}}(\mathbf{x}_i', y_i) \quad \text{s.t.} \quad \boldsymbol{\nu}^* \triangleq \sum_{i=1}^{n} v_i \delta_{\mathbf{x}_i'} = \max_{\boldsymbol{\nu} \in \mathcal{S}_{\boldsymbol{\mu}}} \mathcal{D}(\boldsymbol{\mu}, \boldsymbol{\nu}). \tag{7}$$

The proposed formulation deviates from the conventional minimax formulation for adversarial training [24, 36]. More specifically, it can be regarded as an instance of the more general *bilevel optimization* problem [13, 3]. Feature scattering is effective for adversarial training scenario as there is a requirements of more data [52]. Feature scattering promotes data diversity without drastically altering the structure of the data manifold as in the conventional supervised approach, with label leaking as one manifesting phenomenon. Secondly, the feature matching distance couples the samples within the batch together, therefore the generated adversarial attacks are produced collaboratively by taking the inter-sample relationship into consideration. Thirdly, feature scattering implicitly induces a coupled regularization (detailed below) on model training, leveraging the inter-sample structure for joint regularization.

The proposed approach is equivalent to the minimization of a loss, $\frac{1}{n} \sum_{i=1}^{n} \mathcal{L}_{\boldsymbol{\theta}}(\mathbf{x}_i, y_i) + \lambda \mathcal{R}_{\boldsymbol{\theta}}(\mathbf{x}_1, \cdots, \mathbf{x}_n)$, consisting of the conventional loss $\mathcal{L}_{\boldsymbol{\theta}}(\mathbf{x}_i, y_i)$ on the original data, and a regularization term $\mathcal{R}_{\boldsymbol{\theta}}$ coupled over the inputs. It first highlights the unique property of the proposed feature scattering approach that it induces an effective regularization term that is coupled over *all* inputs, *i.e.*, $\mathcal{R}_{\boldsymbol{\theta}}(\mathbf{x}_1, \cdots, \mathbf{x}_n) \neq \sum_i \mathcal{R}_{\boldsymbol{\theta}}'(\mathbf{x}_i)$. This implies that the model leverages information from all inputs in a joint fashion for learning, offering the opportunity of collaborative regularization leveraging inter-sample relationships. Second, the usage of a function ($\mathcal{D}_{\boldsymbol{\theta}}$) different from $\mathcal{L}_{\boldsymbol{\theta}}$ for inducing $\mathcal{R}_{\boldsymbol{\theta}}$ offers more flexibilities in the effective regularization; moreover, no label information is incorporated in $\mathcal{D}_{\boldsymbol{\theta}}$, thus avoiding potential label leaking as in the conventional case when $\frac{\partial \mathcal{L}_{\boldsymbol{\theta}}(\mathbf{x}_i, y_i)}{\partial \mathbf{x}_i}$ is highly correlated with $y_i$. Finally, in the case when $\mathcal{D}_{\boldsymbol{\theta}}$ is separable over inputs and takes the form of a supervised loss, *e.g.*, $\mathcal{D}_{\boldsymbol{\theta}} \equiv \sum_i \mathcal{L}_{\boldsymbol{\theta}}(\mathbf{x}_i, y_i)$, the proposed approach reduces to the conventional adversarial training setup [24, 36]. The overall procedure for the proposed approach is in Algorithm 1.

---

**Algorithm 1** Feature Scattering-based Adversarial Training

**Input:** dataset $S$, training epochs $K$, batch size $n$, learning rate $\gamma$, budget $\epsilon$, attack iterations $T$
**for** $k = 1$ **to** $K$ **do**
    **for** random batch $\{\mathbf{x}_i, y_i\}_{i=1}^{n} \sim S$ **do**
        **initialization**: $\boldsymbol{\mu} = \sum_i u_i \delta_{\mathbf{x}_i}, \quad \boldsymbol{\nu} = \sum_i v_i \delta_{\mathbf{x}_i'}, \quad \mathbf{x}_i' \sim B(\mathbf{x}_i, \epsilon)$
        **feature scattering** (maximizing feature matching distance $\mathcal{D}$ *w.r.t.* $\boldsymbol{\nu}$):
        **for** $t = 1$ **to** $T$ **do**
          · $\mathbf{x}_i' \leftarrow \mathcal{P}_{\mathcal{S}_{\mathbf{x}}}\big(\mathbf{x}_i' + \epsilon \cdot \text{sign}(\nabla_{\mathbf{x}_i'} \mathcal{D}(\boldsymbol{\mu}, \boldsymbol{\nu}))\big) \quad \forall i = 1, \cdots, n, \quad \boldsymbol{\nu} = \sum_i v_i \delta_{\mathbf{x}_i'}$
        **end for**
        **adversarial training** (updating model parameters):
        · $\boldsymbol{\theta} \leftarrow \boldsymbol{\theta} - \gamma \cdot \frac{1}{n} \sum_{i=1}^{n} \nabla_{\boldsymbol{\theta}} \mathcal{L}(\mathbf{x}_i', y_i; \boldsymbol{\theta})$
    **end for**
**end for**
**Output:** model parameter $\boldsymbol{\theta}$.

---

## 4 Discussions

**Manifold-based Defense [34, 37, 15, 27]**. [34, 37, 27] proposed to defend by projecting the perturbed image onto a proper manifold. [15] used a similar idea of manifold projection but approximated this step with a nearest neighbor search against a web-scale database. Differently, we leverage the manifold in the form of inter-sample relationship for the generation of the perturbations, which induces an implicit regularization of the model when used in the adversarial training framework. While defense in [34, 37, 15, 27] is achieved by *shrinking* the perturbed inputs towards the manifold, we *expand* the manifold using feature scattering to generate perturbed inputs for adversarial training.

**Inter-sample Regularization [70, 30, 39]**. Mixup [70] generates training examples by linear interpolation between pairs of natural examples, thus introducing an linear inductive bias in the vicinity of training samples. Therefore, the model is expected to reduce the amount of undesirable oscillations for off-manifold samples. Logit pairing [30] augments the original training loss with a "pairing" loss, which measures the difference between the logits of clean and adversarial images. The idea is to suppress spurious logits responses using the natural logits as a reference. Similarly, virtual adversarial training [39] proposed a regularization term based on the KL divergence of the prediction probability of original and adversarially perturbed images. In our model, the inter-sample relationship is leveraged for generating the adversarial perturbations, which induces an implicit regularization term in the objective function that is coupled over all input samples.

**Wasserstein GAN and OT-GAN [1, 49, 10]**. Generative Adversarial Networks (GAN) is a family of techniques that learn to capture the data distribution implicitly by generating samples directly [23]. It originally suffers from the issues of instability of training and mode collapsing [23, 1]. OT-related distances [1, 12] have been used for overcoming the difficulties encountered in the original GAN training [1, 49]. This technique has been further extended to generating discrete data such as texts [10]. Different from GANs, which maximizes a discrimination criteria *w.r.t.* the *parameters* of the discriminator for better capturing the data distribution, we maximize a feature matching distance *w.r.t.* the perturbed *inputs* for generating proper training data to improve model robustness.

## 5 Experiments

**Baselines and Implementation Details**. Our implementation is based on PyTorch and the code as well as other related resources are available on the project page.[3] We conduct extensive experiments across several benchmark datasets including CIFAR10 [31], CIFAR100 [31] and SVHN [42]. We use Wide ResNet (WRN-28-10) [68] as the network structure following [36]. We compare the performance of the proposed method with a number of baseline methods, including: *i)* the model trained with standard approach using clean images (`Standard`) [31], *ii)* PGD-based approach from Madry *et al.* (`Madry`) [36], which is one of the most effective defense method [2], *iii)* another recent method performs adversarial training with both image and label adversarial perturbations (`Bilateral`) [61]. For *training*, the initial learning rate $\gamma$ is 0.1 for CIFAR and 0.01 for SVHN. We set the number of epochs the `Standard` and `Madry` methods as 100 with transition epochs as $\{60, 90\}$ as we empirically observed the performance of the trained model stabilized before 100 epochs. The training scheduling of 200 epochs similar to [61] with the same transition epochs used as we empirically observed it helps with the model performance, possibly due to the increased variations of data via feature scattering. We performed standard data augmentation including random crops with 4 pixels of padding and random horizontal flips [31] during training. The perturbation budget of $\epsilon = 8$ is used in training following literature [36]. Label smoothing of 0.5, attack iteration $T=1$ and Sinkhorn algorithm [12] with regularization of 0.01 is used. For *testing*, model robustness is evaluated by approximately computing an upper bound of robustness on the test set, by measuring the accuracy of the model under different adversarial attacks, including white-box FGSM [24], PGD [36], CW [8] (CW-loss [8] within the PGD framework) attacks and variants of black-box attacks.

### 5.1 Visual Classification Performance Under White-box Attacks

**CIFAR10**. We conduct experiments on CIFAR10 [31], which is a popular dataset that is widely use in adversarial training literature [36, 61] with 10 classes, 5K training images per class and 10K test images. We report the accuracy on the original test images (Clean) and under PGD and CW attack with $T$ iterations (PGD$T$ and CW$T$) [36, 8]. The evaluation results are summarized in Table 1. It is observed `Standard` model fails drastically under different white-box attacks. `Madry` method improves the model robustness significantly over the `Standard` model. Under the standard PGD20 attack, it achieves 44.9% accuracy. The `Bilateral` approach further boosts the performance to 57.5%. The proposed approach outperforms both methods by a large margin, improving over `Madry` by 25.6%, and is 13.0% better than `Bilateral`, achieving 70.5% accuracy under the standard 20 steps PGD attack. Similar patten has been observed for CW metric.

We further evaluate model robustness against PGD attacker under different attack budgets with a fixed attack step of 20, with the results shown in Figure 3 (a). It is observed that the performance of `Standard` model drops quickly as the attack budget increases. The `Madry` model [36] improves the model robustness significantly across a wide range of attack budgets. The `Proposed` approach

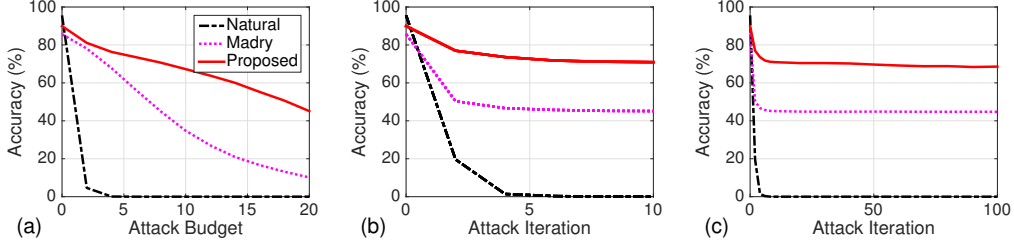

Figure 3: **Model performance** under PGD attack with different (a) attack budgets (b) attack iterations. `Madry` and `Proposed` models are trained with the attack iteration of 7 and 1 respectively.

| Models | Clean | Accuracy under White-box Attack ($\epsilon = 8$) | | | | | | | | |
|---|---|---|---|---|---|---|---|---|---|---|
| | | FGSM | PGD10 | PGD20 | PGD40 | PGD100 | CW10 | CW20 | CW40 | CW100 |
| `Standard` | **95.6** | 36.9 | 0.0 | 0.0 | 0.0 | 0.0 | 0.0 | 0.0 | 0.0 | 0.0 |
| `Madry` | 85.7 | 54.9 | 45.1 | 44.9 | 44.8 | 44.8 | 45.9 | 45.7 | 45.6 | 45.4 |
| `Bilateral` | 91.2 | 70.7 | – | 57.5 | – | 55.2 | – | 56.2 | – | 53.8 |
| `Proposed` | 90.0 | **78.4** | **70.9** | **70.5** | **70.3** | **68.6** | **62.6** | **62.4** | **62.1** | **60.6** |

Table 1: Accuracy comparison of the `Proposed` approach with `Standard`, `Madry` [36] and `Bilateral` [61] methods on CIFAR10 under different threat models.

further boosts the performance over the `Madry` model [36] by a large margin under different attack budgets. We also conduct experiments using PGD attacker with different attack iterations with a fixed attack budget of 8, with the results shown in Figure 3 (b-c) and also Table 1. It is observed that both `Madry` [36] and `Proposed` can maintain a fairly stable performance when the number of attack iterations is increased. Notably, the proposed approach consistently outperforms the `Madry` [36] model across a wide range of attack iterations. From Table 1, it is also observed that the `Proposed` approach also outperforms `Bilateral` [61] under all variants of PGD and CW attacks. We will use a PGD/CW attackers with $\epsilon$=8 and attack step 20 and 100 in the sequel as part of the threat models.

| Models | Clean | White-box Attack ($\epsilon$=8) | | | | |
|---|---|---|---|---|---|---|
| | | FGSM | PGD20 | PGD100 | CW20 | CW100 |
| `Standard` | **97.2** | 53.0 | 0.3 | 0.1 | 0.3 | 0.1 |
| `Madry` | 93.9 | 68.4 | 47.9 | 46.0 | 48.7 | 47.3 |
| `Bilateral` | 94.1 | 69.8 | 53.9 | 50.3 | – | 48.9 |
| `Proposed` | 96.2 | **83.5** | **62.9** | **52.0** | **61.3** | **50.8** |

| Models | Clean | White-box Attack ($\epsilon$=8) | | | | |
|---|---|---|---|---|---|---|
| | | FGSM | PGD20 | PGD100 | CW20 | CW100 |
| `Standard` | **79.0** | 10.0 | 0.0 | 0.0 | 0.0 | 0.0 |
| `Madry` | 59.9 | 28.5 | 22.6 | 22.3 | 23.2 | 23.0 |
| `Bilateral` | 68.2 | 60.8 | 26.7 | 25.3 | – | 22.1 |
| `Proposed` | 73.9 | **61.0** | **47.2** | **46.2** | **34.6** | **30.6** |

Table 2: Accuracy comparison on (a) SVHN and (b) CIFAR100.

**SVHN**. We further report results on the SVHN dataset [42]. SVHN is a 10-way house number classification dataset, with 73257 training images and 26032 test images. The additional training images are not used in experiment. The results are summarized in Table 2(a). Experimental results show that the proposed method achieves the best clean accuracy among all three robust models and outperforms other method with a clear margin under both PGD and CW attacks with different number of attack iterations, demonstrating the effectiveness of the proposed approach.

**CIFAR100**. We also conduct experiments on CIFAR100 dataset, with 100 classes, 50K training and 10K test images [31]. Note that this dataset is more challenging than CIFAR10 as the number of training images per class is ten times smaller than that of CIFAR10. As shown by the results in Table 2(b), the proposed approach outperforms all baseline methods significantly, which is about 20% better than `Madry` [36] and `Bilateral` [61] under PGD attack and about 10% better under CW attack. The superior performance of the proposed approach on this data set further demonstrates the importance of leveraging inter-sample structure for learning [69].

## 5.2 Ablation Studies

We investigate the impacts of algorithmic components and more results are in the supplementary file.

**The Importance of Feature Scattering**. We empirically verify the effectiveness of feature scattering, by comparing the performances of models trained using different perturbation schemes: *i)* `Random`: a natural baseline approach that randomly perturb each sample within the epsilon neighborhood; *ii)* `Supervised`: perturbation generated using ground-truth label in a supervised fashion; *iii)* `FeaScatter`: perturbation generated using the proposed feature scattering method. All other hyper-parameters are kept exactly the same other than the perturbation scheme used. The results are summarized in Table 3(a). It is evident that the proposed feature scattering (`FeaScatter`) approach

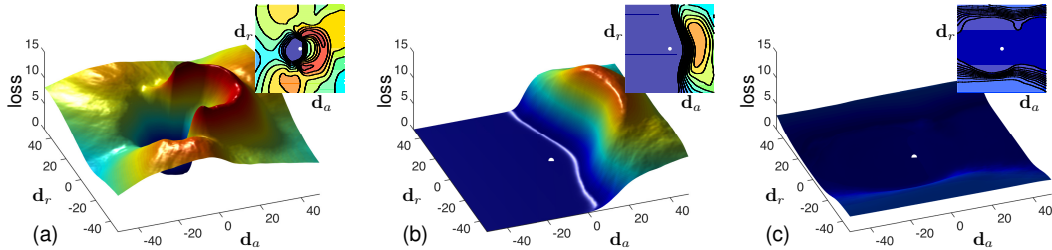

Figure 4: **Loss surface visualization** in the vicinity of a natural image along adversarial direction ($\mathbf{d}_a$) and direction of a Rademacher vector ($\mathbf{d}_r$) for (a) `Standard` (b) `Madry` (c) `Proposed` models.

outperforms both `Random` and `Supervised` methods, demonstrating its effectiveness. Furthermore, as it is the major component that is difference from the conventional adversarial training pipeline, this result suggests that feature scattering is the main contributor to the improved adversarial robustness.

| Perturb | Clean | White-box Attack ($\epsilon=8$) | | | | | Match | Clean | White-box Attack ($\epsilon=8$) | | | | |
|---|---|---|---|---|---|---|---|---|---|---|---|---|---|
| | | FGSM | PGD20 | PGD100 | CW20 | CW100 | | | FGSM | PGD20 | PGD100 | CW20 | CW100 |
| `Random` | 95.3 | 75.7 | 29.9 | 18.3 | 34.7 | 26.2 | `Uniform` | 90.0 | 71.0 | 57.1 | 54.7 | 53.2 | 51.4 |
| `Supervised` | 86.9 | 64.4 | 56.0 | 54.5 | 51.2 | 50.3 | `Identity` | 87.4 | 66.3 | 57.5 | 56.0 | 52.4 | 50.6 |
| `FeaScatter` | 90.0 | 78.4 | 70.5 | 68.6 | 62.4 | 60.6 | `OT` | 90.0 | 78.4 | 70.5 | 68.6 | 62.4 | 60.6 |

Table 3: (a) Importance of feature-scattering. (b) Impacts of different matching schemes.

**The Role of Matching**. We further investigate the role of matching schemes within the feature scattering component by comparing several different schemes: *i)* `Uniform` matching, which matches each clean sample uniformly with all perturbed samples in the batch; *ii)* `Identity` matching, which matches each clean sample to its perturbed sample only; *iii)* `OT-matching`: the proposed approach that assigns soft matches between the clean samples and perturbed samples according to the optimization criteria. The results are summarized in Table 3(b). It is observed all variants of matching schemes lead to performances that are on par or better than state-of-the-art methods, implying that the proposed framework is effective in general. Notably, `OT-matching` leads to the best results, suggesting the importance of the proper matching for feature scattering.

**The Impact of OT-Solvers**. Exact minimization of Eqn.(4) over $\mathbf{T}$ is intractable in general [1, 49, 21, 12]. Here we compare two practical solvers, the `Sinkhorn` algorithm [12] and the Inexact Proximal point method for Optimal Transport (`IPOT`) algorithm [66]. More details on them can be found in the supplementary file and [12, 66, 45]. The results are summarized in Table 4. It is shown that different instantiations of the proposed approach with different OT-solvers lead to comparable performances, implying that the proposed approach is effective in general regardless of the choice of OT-solvers.

| OT-solver | CIFAR10 | | | | | | SVHN | | | | | | CIFAR100 | | | | | |
|---|---|---|---|---|---|---|---|---|---|---|---|---|---|---|---|---|---|---|
| | Clean | FGSM | PGD20 | PGD100 | CW20 | CW100 | Clean | FGSM | PGD20 | PGD100 | CW20 | CW100 | Clean | FGSM | PGD20 | PGD100 | CW20 | CW100 |
| `Sinkhorn` | 90.0 | 78.4 | 70.5 | 68.6 | 62.4 | 60.6 | 96.2 | 83.5 | 62.9 | 52.0 | 61.3 | 50.8 | 73.9 | 61.0 | 47.2 | 46.2 | 34.6 | 30.6 |
| `IPOT` | 89.9 | 77.9 | 69.9 | 67.3 | 59.6 | 56.9 | 96.0 | 82.6 | 60.0 | 49.3 | 57.8 | 48.4 | 74.2 | 67.3 | 47.5 | 46.3 | 32.0 | 29.3 |

Table 4: Impacts of OT-solvers. The proposed approach performs well with different OT-solvers.

## 5.3 Performance under Black-box Attack

To further verify if a degenerate minimum is obtained, we evaluate the robustness of the model trained with the proposed approach *w.r.t. black-box* attacks (B-Attack) following [58]. Two different models are used for generating

| B-Attack | PGD20 | PGD100 | CW20 | CW100 |
|---|---|---|---|---|
| `Undefended` | 89.0 | 88.7 | 88.9 | 88.8 |
| `Siamese` | 81.6 | 81.0 | 80.3 | 79.8 |

test time attacks: *i)* `Undefended`: undefended model trained using `Standard` approach, *ii)* `Siamese`: a robust model from another training session using the proposed approach. As demonstrated by the results in the table on the right, the model trained with the proposed approach is robust against different types of black-box attacks, verifying that a non-degenerate solution is learned [58].

Finally, we visualize in Figure 4 the loss surfaces of different models as another level of comparison.

## 6 Conclusion

We present a feature scattering-based adversarial training method in this paper. The proposed approach distinguish itself from others by using an unsupervised feature-scattering approach for generating adversarial training images, which leverages the inter-sample relationship for collaborative perturbation generation. We show that a coupled regularization term is induced from feature scattering for adversarial training and empirically demonstrate the effectiveness of the proposed approach through extensive experiments on benchmark datasets.

## Footnotes

[2]The two discrete distributions could be of different dimensions; here we present the exposition assuming the same dimensionality to avoid notion clutter.

[3]https://sites.google.com/site/hczhang1/projects/feature_scattering

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
