[Supplementary Material]

# Defense Against Adversarial Attacks Using Feature Scattering-based Adversarial Training – Supplementary File

**Haichao Zhang**[*]          **Jianyu Wang**
Horizon Robotics          Baidu Research
hczhang1@gmail.com   wjyouch@gmail.com

## 1   OT Solver Details

We have used the OT distance as in Eqn.(4) for feature scattering (Eqn.(6)). However, computing the OT distance exactly (or equivalently, the exact minimization over $\mathbf{T}$) is in general computational intractable [1, 3, 13]. In this work, we use two existing methods from the literature. the Sinkhorn algorithm [3] and the Inexact Proximal point method for Optimal Transport (IPOT) algorithm [14], which have been adopted for different applications [8, 4, 2, 11, 7].

### 1.1   The Sinkhorn algorithm [3]

Cuturi proposed the seminal work on introducing an entropy regularization into the original OT formulation and using Sinkhorn's matrix scaling algorithm to solve the regularized OT problem [3]. Specifically, the Sinkhorn algorithm tries to solve the entropy regularized optimization problem following [3]:

$$\mathcal{D}(\boldsymbol{\mu}, \boldsymbol{\nu}) = \min_{\mathbf{T} \in \Pi(\mathbf{u}, \mathbf{v})} \langle \mathbf{T}, \mathbf{C} \rangle - \lambda h(\mathbf{T}), \tag{1}$$

where $h(\mathbf{T}) = -\sum_{i,j} \mathbf{T}_{ij}(\log(\mathbf{T}_{ij}) - 1)$ denotes an entropic regularization term and $\lambda$ is the regularization weight. We will use $\mathbf{f}_i$ in the sequel to denote a general feature vector in order to present the algorithm in the general form. In our work, $\mathbf{f}_i = f_{\boldsymbol{\theta}}(\mathbf{x}_i)$.

---

**Algorithm 1** Sinkhorn algorithm [3]

---

1: **Input:** Feature vectors $\mathbf{F} = \{\mathbf{f}_i\}_1^n$, $\mathbf{F}' = \{\mathbf{f}'_j\}_1^n$ , $\tau$
2: $\boldsymbol{l} = \frac{1}{n}\mathbf{1}$      // $\mathbf{1}$ denotes an all-ones vector of dimension $n$
3: $\mathbf{C}_{ij} = c(\mathbf{f}_i, \mathbf{f}'_j)$, $\mathbf{K}_{ij} = e^{-\frac{\mathbf{C}_{ij}}{\tau}}$
4: **for** $t = 1, \cdots, T$ **do**
5:     $\boldsymbol{r} = \frac{1}{\mathbf{K}^T \boldsymbol{l}}, \boldsymbol{l} = \frac{1}{\mathbf{K}\boldsymbol{r}}$
6: **end for**
7: $\mathbf{T}^* = \text{diag}(\boldsymbol{l})\mathbf{K}\text{diag}(\boldsymbol{r})$
8: **Output:** the OT-distance $\langle \mathbf{T}^*, \mathbf{C} \rangle$.

---

### 1.2   The IPOT algorithm [6]

An Inexact Proximal point method for Optimal Transport (IPOT) is developed in [14] for computing the OT distance. IPOT provides a solution to the original OT problem specified in Eqn.(4). Specifically, IPOT iteratively solves the following optimization problem using the proximal point method [6]:

$$\mathbf{T}^{(t+1)} = \arg\min_{\mathbf{T} \in \Pi(\mathbf{u}, \mathbf{v})} \left\{ \langle \mathbf{T}, \mathbf{C} \rangle + \beta \cdot \mathcal{E}(\mathbf{T}, \mathbf{T}^{(t)}) \right\}, \tag{2}$$

---

[*]Work done while with Baidu Research.

where the proximity metric term $\mathcal{E}(\mathbf{T}, \mathbf{T}^{(t)})$ penalizes solutions that are too distant from the latest approximation, and $\frac{1}{\beta}$ is understood as the generalized stepsize. This renders a tractable iterative scheme towards the exact OT solution. In this work, we employ the generalized KL Bregman divergence $\mathcal{E}(\mathbf{T}, \mathbf{T}^{(t)}) = \sum_{i,j} \mathbf{T}_{ij} \log \frac{\mathbf{T}_{ij}}{\mathbf{T}_{ij}^{(t)}} - \sum_{i,j} \mathbf{T}_{ij} + \sum_{i,j} \mathbf{T}_{ij}^{(t)}$ as the proximity metric. More detailed procedures for IPOT are summarized in Algorithm 2.

---

**Algorithm 2** IPOT algorithm [6]

---
1: **Input:** Feature vectors $\mathbf{F} = \{\mathbf{f}_i\}_1^n$, $\mathbf{F}' = \{\mathbf{f}'_j\}_1^n$ and generalized step size $1/\beta$
2: $\boldsymbol{r} = \frac{1}{n}\mathbf{1}$, $\mathbf{T}^{(1)} = \mathbf{1}\mathbf{1}^\top$ // $\mathbf{1}$ denotes an all-ones vector of dimension $n$
3: $\mathbf{C}_{ij} = c(\mathbf{f}_i, \mathbf{f}'_j)$, $\mathbf{K}_{ij} = \mathrm{e}^{-\frac{\mathbf{C}_{ij}}{\beta}}$
4: **for** $t = 1, \cdots, T$ **do**
5: $\quad \mathbf{Q} = \mathbf{K} \odot \mathbf{T}^{(t)}$ // $\odot$ is Hadamard product
6: $\quad$ **for** $k = 1, \cdots K$ **do**
7: $\quad\quad \boldsymbol{l} = \frac{1}{n\mathbf{Q}\boldsymbol{r}}, \boldsymbol{r} = \frac{1}{n\mathbf{Q}^T\boldsymbol{l}}$
8: $\quad$ **end for**
9: $\quad \mathbf{T}^{(t+1)} = \mathrm{diag}(\boldsymbol{l})\mathbf{Q}\mathrm{diag}(\boldsymbol{r})$
10: **end for**
11: **Output:** the OT-distance $\langle \mathbf{T}^{(T)}, \mathbf{C} \rangle$.

---

## 2 More Results

### 2.1 Results against Stronger White-box Attacks

We further conduct evaluations against stronger white-box attacks. Specifically, we increase the attack iterations and run the evaluations 5 times with different random starts for each test example and report the lowest performance as in the Table 1. It can be observed that our model still achieves competitive performance under these strong attacks.

| Attack Iteration | PGD500 | PGD1000 | CW500 | CW1000 |
|---|---|---|---|---|
| min over 5 runs | 64.5 | 64.2 | 56.8 | 56.6 |

Table 1: Accuracy results on CIFAR10 under stronger white-box attacks.

### 2.2 Results against Gradient-free Attacks

We evaluated our model using a gradient-free black-box attack based local search [5] and SPSA [9, 12]. The performance of our model under local-search attack is 89.9%, and the performance under SPSA attack [9] is 88.8%. These results further assuring the absence of gradient masking.

| label smoothing $\Delta$ | Clean | White-box Attack ($\epsilon = 8$) | | | | |
|---|---|---|---|---|---|---|
| | | FGSM | PGD20 | PGD100 | CW20 | CW100 |
| 0.3 | 90.3 | 77.1 | 66.0 | 61.8 | 62.4 | 59.4 |
| 0.4 | 90.3 | 77.0 | 64.0 | 58.9 | 61.2 | 58.0 |
| 0.5 | 90.1 | 78.2 | 70.7 | 68.4 | 62.8 | 60.0 |
| 0.6 | 89.8 | 78.0 | 71.7 | 70.1 | 62.5 | 60.5 |
| 0.7 | 90.3 | 77.3 | 72.4 | 71.2 | 67.4 | 65.7 |
| 0.8 | 90.5 | 74.0 | 70.3 | 69.7 | 66.5 | 65.6 |

Table 2: Accuracy results on CIFAR10 under different label-smoothing parameters.

### 2.3 Label Smoothing Parameter

Label smoothing regularization [10] is used which models the smoothed target distribution as:

$$q(c) = (1 - \Delta)\delta_{c,y} + \frac{\Delta}{C - 1} \tag{3}$$

where $y$ is the ground-truth label, $C$ denotes the total number of classes and $\Delta$ the smoothing parameter. Label smoothing is necessary for our model. Our model (with 1-step adversary) achieves compromised results without it, compared to standard PGD adversarial training (e.g. Madry) with 7-steps adversaries. This is an expected result as feature scattering makes the feature distributions more diffused (Figure 2), thus the corresponding label should ideally be "diffused" as well. We empirically show that the proposed approach is not very sensitive to the label smoothing parameter and performs reasonably well across a wide range of smoothing parameter values. The results are shown in Table 2. It is observed that the proposed approach performs well within a wide range of values for the label smoothing parameter, all outperforming other baseline methods.

## 2.4 OT-Parameters

The performance of the proposed approach is relatively stable within a wide range of value for the parameter $\tau$ in the Sinkhorn algorithm 1, as can be observed from the results in Table 3. Results for IPOT algorithm is presented in Table 4. It is observed that the proposed approach performs comparably well under different parameter choices.

| $\tau$ | Clean | White-box Attack ($\epsilon=8$) | | | | |
|---|---|---|---|---|---|---|
| | | FGSM | PGD20 | PGD100 | CW20 | CW100 |
| 0.001 | 90.4 | 77.4 | 66.9 | 64.6 | 60.8 | 58.8 |
| 0.005 | 90.1 | 77.7 | 68.7 | 66.5 | 60.3 | 58.5 |
| 0.01 | 90.0 | 78.4 | 70.5 | 68.6 | 62.4 | 60.6 |
| 0.05 | 89.5 | 76.6 | 67.3 | 65.3 | 59.4 | 57.6 |
| 0.5 | 90.3 | 76.5 | 65.6 | 63.6 | 60.2 | 58.2 |

Table 3: Accuracy results on CIFAR10 using Sinkhorn using different parameters.

| $\beta$ | Clean | White-box Attack ($\epsilon=8$) | | | | |
|---|---|---|---|---|---|---|
| | | FGSM | PGD20 | PGD100 | CW20 | CW100 |
| 0.5 | 90.1 | 78.3 | 70.7 | 68.2 | 60.5 | 57.9 |
| 1 | 89.9 | 77.9 | 69.9 | 67.3 | 59.6 | 56.9 |

Table 4: Accuracy results on CIFAR10 using IPOT using different parameters.

## 2.5 Loss Plots

The loss achieved with PGD adversary against our model increases in a fairly consistent way and plateaus rapidly for 20 different runs with random starts, with relatively small variance (Figure 1).

Figure 1: **Loss function** value over PGD iterations for 20 random starts on random examples from CIFAR10 test set.

## 3 Visualization of Feature Embeddings

We visualize the image features in Figure 2. The feature clusterings using the model trained with the `Standard` approach is shown in the *top* row of Figure 2. It is observed that supervised (label-guided) attacks have a strong biased movement directions (Figure 2 (b)), *i.e.*, the data point is guided towards the decision boundary to flip the decision, therefore leading to mixed labels within each cluster

for features from the `Standard` model. The proposed feature approach is less decision-boundary-oriented, as it is unsupervised in nature, thus potentially covers more space around the manifold, as demonstrated by the less structured cluster on `Standard` model by feature scattering (Figure 2 (c)).

The feature clusterings using the model trained with the `Proposed` approach is shown in the *bottom* row of Figure 2. It is observed that the model is robust against supervised attack (Figure 2 (e)) as each cluster largely maintains its label identity and suffers less from label mixing as for the `Standard` model (Figure 2 (b)). Similarly, feature scattering mainly leads to a diffusion of the clusters without severe mixing (Figure 2 (f) ) as for the `Standard` model (Figure 2) (c)).

Figure 2: **Visualization of features**. Left: clean images; middle: standard (supervised) attack; right: feature scattering. Top: features extracted from a model trained with `Standard` method. Bottom: features extracted with a model train with the `Proposed` method.