[Reviews · NeurIPS 2019]

Reviewer 1



To my knowledge, the idea of coupling perturbations across examples is a new idea, and worthy of additional exploration. This paper makes a nice contribution in that direction. The idea, and algorithmic instantiation, both seem well-done. The paper claims (significantly) state-of-the-art results across a variety of tasks. Currently, the adversarial evaluation seems suspect, but if these holes are addressed, this paper would be a strong contribution in originality, quality, and significance. (I remain unclear whether the proposed algorithm ought to be an improvement over standard adversarial training, and thus, the empirical results are particularly important here.) My major concerns are whether gradient masking is present in the model, and whether it is tested against the strongest possible attacks. I will focus on CIFAR-10 at eps=8, as this is by far the most competitive benchmark. First, it is suspicious that the black-box attack (transferring from an undefended model) does better than the white-box attack. Against most adversarially robust models, black box attacks are extremely weak. E.g. Madry reports 86% accuracy when transferring from an undefended model. Second, the large gap between PGD and the CW-variant (the paper says this is PGD using the CW loss) is suspicious. If I understand correctly, the only difference is a cross-entropy loss vs. a margin loss. In typical models, e.g. Madry, these converge to similar values by 100 iterations, but here there is an 8% gap (68.6 vs 60.6). Several comments on clarity, and miscellaneous questions: I found the description of Algorithm 1 unclear. For estimating grad_x’ D(mu, nu), my understanding is that the algorithm first estimates the transport matrix T (using e.g. Sinkhorn), and then computes the gradient treating T as fixed, ignoring the dependence of T on x’. On first read, it was unclear how this gradient was estimated, and this seemed the most sensible to me, but please correct me if this interpretation is mistaken. How are u_i and v_i defined? I couldn’t find this in the paper. The natural choice seems to be 1/N (uniform over the dataset), but why introduce u_i and v_i in this case? e.g. the description in equation (7) would be more straightforward. Minor: I would mention Sinkhorn and IPOT sooner. Otherwise the reader is wondering how the max in Eq. 6 is solved, until the end of the experimental section. Minor: Feature scattering seems to combine two distinct ideas: first, using an unsupervised adversary operating on distances between activations (rather than labels) and second, coupling perturbations across examples. It would be nice to decouple the impacts of these — looking at the Identity matching scheme is nice (as this isolates the first idea without the second), and the comparison here could be developed further. Why pick label smoothing lambda=0.5? In the supplement, lambda=0.8 seems significantly better (65% against strongest adversary vs. 60%). It’s nice that only 1 attack iteration is necessary for the reported results. I think this is worth emphasizing earlier. Finally - I’m glad to see the authors note that they intend to open source their model and evaluation code. I believe this is a great practice for the adversarial robustness community. __________________ Update: I have changed my score from a 5 to an 7, largely in response to the noticed bug regarding reported black-box numbers in fact corresponding to a white-box evaluation (as well as additional convergence plots, and stronger attacks). With the reconciled results, the paper proposes a promising idea (computing perturbations which are coupled across examples), and a significant improvement over SOTA on a competitive benchmark (CIFAR-10 at eps=8, white-box). I am somewhat hesitant after the initial mistake. In particular, this seems like a sanity check which the authors should have noticed before the submission, and the fact that it was unnoticed is somewhat concerning. The fact that the code is being open-sourced, and so that it will be relatively easy for the community to verify the claims made in the paper is a significant contributing factor to my updated score, and not dwelling too much on this oversight. I would also encourage the authors to include the additional adversarial evaluations (with fixes as suggested by other reviewers) and ablation studies (over label smoothing parameter) in the final version.

Reviewer 2



Update after reading authors response: I changed my score to accept after reading author's response because authors addressed most of the comments from reviewers and they also explain that initial black box results [which created impression of gradient masking] were erroneously entered. Still my concern about gradient free attack not fully addressed. Authors did provide result on gradient free attack, however chosen method (http://www.shivakasiviswanathan.com/CVPR17W.pdf) seem to be weak attack. So I would highly encourage authors to perform experiments using few additional gradient free attacks (ex: https://arxiv.org/pdf/1802.05666.pdf ) and report it in the final version of the paper. ----------------------------------------------- Original review: Originality: The paper proposes a novel technique to improve model robustness against L-infinity adversarial attacks. The technique takes into account similarity between features in the minibatch. Quality: There are several issues with the paper which makes me lean towards rejection: 1) One big question which seems to be not address is computational complexity of the proposed method. With the same number of inner iterations T, the proposed method requires more compute compared to PGD adversarial training. So assuming that number of iterations T were the same in the experiments, proposed method is actually given some advantage in terms of extra compute. 2) I could not find information about number of iterations T of inner optimization which was used in experiments. I assume that it was the same for PGD adversarial training and feature scattering, but it’s not clear from the text. 3) One of the issues which feature scattering claim to be addressing is label leaking. However as authors mentioned it could be addressed by other means, like guessing labels. So I wonder how proposed method would compare with PGD adversarial training which does not use true labels. 4) There are issues with evaluation of proposed defenses: 4a) According to table 3, proposed method reaches 68.6 accuracy against PGD100 attack and 60.6 accuracy against CW100 attack in white box case. At the same time according to table in the section 5.3, it has lower accuracy against same attacks in black box case. The fact that white box accuracy is lower compared to black box accuracy is usually an indication of gradient masking. 4b) Authors only limit their evaluation to no more than 100 iterations of PGD attack. Also it’s not clear whether they do random restarts and how many. And from the data tables is could be observed that accuracy is decreasing with increase of number of iterations. This means that attack which was used for evaluation might be too weak. 4c) No attempt to use gradient free attacks to evaluate robustness. Clarity: Paper is reasonably well structured. The section about feature scattering is somewhat harder to read. I would encourage authors to expand their intuitive explanation on what happening during feature scattering. Also it would be useful to include intuitive explanation and/or illustration on what happening in feature scattering adversarial training. Significance: If all issues with evaluation are addressed and method still shows improvement over baseline then I would say it has moderate significant: it does not solve the problem of adversarial examples completely, but it does provide an interesting idea and improved metrics.

Reviewer 3



****** Update ******* Like other reviewers, I'm happy to see that there was a good explanation for why the black-box setting was broken. There are two additional points, I'd like to make: 1) The authors either explain the multiple restarts setting wrongly or apply it wrongly. Instead of running 5 separate evaluations and picking the worst (min of mean of accuracy under attack), they should repeat for each example in the dataset the attack 5 times and take the worst (mean of min). This is really important and the authors to fix this and add these results to the paper. 2) I encourage the authors to share their model as quickly as possible and before the conference. This will allow the community to make sure that the authors claims are correct. If they'd rather have this evaluation made in private before fully releasing the model, I'm happy for the AC to transmit my contact details. All in all, I believe the current score is fair if the authors follow-up on their promise to open-source their code (and hopefully model). *********************** The authors introduce a feature scattering-based adversarial training approach (based on solving an optimal transport problem). The main motivation is avoid label leaking. Hence the authors also use label smoothing. Overall, the paper is well written. The motivation is sound and the evaluation seems appropriate. In any case, the results are very impressive (beating the previous state-of-the-art method by 4% in absolute terms on CIFAR-10 with a similar evaluation - see TRADES: TRadeoff-inspired Adversarial DEfense via Surrogate-loss minimization by Zhang et al.). As always, it is very hard to judge whether the evaluation is done correctly and I would urge the authors to release their models if possible. Pros: * SOTA results. * Novel method to generate adversarial examples. Cons: * Some details are missing (e.g., better study of label smoothing). * The black-box attack is stronger than CW100 which is suspicious. Details: 1) Label leaking seems to be the core of the problem. The authors should define it earlier in the introduction. 2) The transport cost is defined by the cosine distance in logit-space. Can the author motivate this choice? 3) The batch size is likely to be an important factor. Can the author provide results with varying batch sizes? 4) Label smoothing is very large (0.5). On CIFAR-10, I haven't seen such a large smoothing applied to adversarial training. The authors should analyze the effect of such smoothing (or consider removing it). In particular could the author add rows corresponding to 0.0, 0.1 and 0.2 to Table 1 in the supplementary material and also evaluate standard adversarial training (Madry et al.) with such smoothing applied. 5) The table in Section 5.3 (black-box attack) is slightly at odds with the other results. In particular, the examples found using the undefended network seem to transfer extremely well (even beating CW100 on the original model) yielding an accuracy under attack of 59.7% (while CW100 yields 60.6%). Maybe I misunderstood what the authors meant by black-box attack. Minor: a) In Table 1, the notation 0.0 and 0.00 is inconsistent.

[Author Response · NeurIPS 2019]

We thank all the reviewers for recognizing the contribution of our work and providing their valuable comments. We address the questions below and will release the trained model together with code as suggested by reviewers to contribute to the community.

**Shared comment on gradient-masking**. We first address this by conducting analysis following reviewers' suggestions from four complementary perspectives: §1 black-box attack, §2 stronger white-box attack, §3 loss variations and §4 gradient-free attack. The results have verified that *the improved robustness is indeed due to model improvement instead of gradient masking*.

**§1 Results under black-box attacks**. For the black-box attacks, the results for PDG100, CW20 and CW100 in the original submission are incorrect due to our own fault: *they are results under white-box attacks*. We spot this error after sub-mission deadline but were unable to upload the correct version. *We sincerely apologize for this misleading mistake and any additional efforts required from all the reviewers because of it*. The correct black-box results are shown in the table above. This black-box results, together with the white-box results (Table 1 in the main paper and §2 below) suggest that gradient masking is not present in the model and the improved robustness is indeed due to the inherent improvement of the model itself.

| B-Attack | PDG20 | PDG100 | CW20 | CW100 |
|---|---|---|---|---|
| Undefended | 89.0 | 88.7 | 88.9 | 88.8 |
| Siamese | 81.6 | 81.0 | 80.3 | 79.8 |

**§2 Results against stronger white-box attacks**. We have used random starts for all the white-box evaluations here as well as in the main paper. Here we run the evaluations 5 times and report the lowest performance and $(\max-\min)$ as in the table on the right. The accuracy of our model under very strong white-box attacks still outperforms Madry model (44.8/45.4) by a large margin.

| Attack Iteration | PGD500 | PGD1000 | CW500 | CW1000 |
|---|---|---|---|---|
| min over 5 runs | 66.8 | 66.4 | 59.0 | 58.8 |
| max–min | 0.3 | 0.3 | 0.3 | 0.2 |

**§3 Loss plots**. The loss achieved with PGD adversary against our model increases in a fairly consistent way and plateaus rapidly for 20 different runs with random starts, with relatively small variance (right).

**§4 Results against gradient-free attacks**. We evaluated our model using a gradient-free black-box attack based on greedy local search,[1] and result is 89.9, further assuring the absence of gradient masking.

*To Reviewer 1*:

**Clarity and miscellaneous comments**. *i)* Your understanding of Alg 1 is correct. We will add more explanations on this for clarity. *ii)* $\boldsymbol{\mu}=\{u_i\}$ represents the "empirical probability vector" over the support of $\{\mathbf{x}_i\}$. They are set to be uniform as shown in the Sinkhorn and IPOT algorithms in supplementary file, in the absence of prior knowledge. We have used this representation to make the presentation general enough to incorporate prior knowledge when available. We will make this clear and mention the Sinkhorn and IPOT sooner in revision. *iii)* Label smoothing parameter 0.5 is set without hyper-parameter tuning. The investigation on smoothing parameter was done afterwards. We didn't aim to achieve the best performance by hyper-parameter search but instead to show the general applicability of our approach under a broad range of hyper-parameter settings. *iv)* The gap between PGD and the CW-variant is smaller under stronger attacks (§2) and we attribute the remaining gap to the nature of our model, where a one step unsupervised adversary is used for training, different from the multi-step supervised adversary typically used in Madry model. *v)* "Emphasizing 1 attack iteration earlier" is a great suggestion. We will update this to avoid confusions as happened to **Reviewer 2**.

**Disentangling of distance and coupling**. Thanks for the great suggestion. We have preliminarily investigated the disentangling of distance and inter-sample coupling in our main paper as you have already noticed in Sec. 5.2 using the `identity matching`. Further investigation on it (esp. the coupling) as suggested by the reviewer is interesting and we plan to work on it as our next steps.

*To Reviewer 2*:

**Computational concern**. The number of iterations $T=1$ is used for our model as mentioned in `line 225`. We apologize for the confusion and will make it more clear as also suggested by **Reviewer 1**. Given that $T$ is typically set to 7 in conventional PGD adversarial training (e.g. Madry), our approach does not take advantage of extra computation compared to conventional PGD training.

**Random targeted baseline**. We have experimented with random-targeted adversarial training as suggested by the reviewer. It achieves accuracy of 49.9/48.5 under PGD100/CW100 (`min` over 5 runs), outperformed by our model with a large margin (§2).

**Understanding of feature-scattering**. Conventional adversarial examples are *decision boundary oriented* (Fig.2), making the effective manifold for training deviate from the original due to *tilting* and *shrinking*, hindering performance (`line 36-52`), with label leaking as one manifesting phenomenon. Feature scattering is *inter-sample structure* oriented and promotes data diversity without drastically altering the structure of the manifold. We plan to conduct rigorous theoretical analysis of the proposed model as next step.

*To Reviewer 3*:

**Batch size**. As shown in the table on the right, larger batch size leads to better performance as it facilitates feature matching. Batch size of 60 is used for our model in the paper. Batch sizes larger than 60 lead to similar results. This observation is similar to other applications with embedded OT matching such as OT-GAN [48].

| batch size | 40 | 50 | 60 | 70 | 80 |
|---|---|---|---|---|---|
| PDG100 | 58.7 | 62.7 | 68.4 | 68.2 | 68.4 |

**Label smoothing**. Label smoothing is necessary for our model. Our model (with 1-step adversary) achieves compromised results without it, compared to standard PGD adversarial training (e.g. Madry) with 7-steps adversaries. This is an expected result as feature scattering makes the feature distributions more diffused (see Fig 1 in supplementary file), thus the corresponding label should ideally be "diffused" as well, in a spirit similar to *mixup* [70], which is achieved with label smoothing approximately in this work. Better schemes for joint treatment of feature and label scattering is an interesting topic and is left as our future work.

| smooth para. | 0 | 0.1 | 0.2 | 0.3 | 0.4 | 0.5 | 0.6 | 0.7 | 0.8 |
|---|---|---|---|---|---|---|---|---|---|
| Madry | 44.8 | 46.4 | 46.7 | 46.1 | 46.2 | 46.1 | 45.6 | 46.8 | 47.4 |
| Ours | 35.3 | 56.2 | 59.2 | 61.8 | 58.9 | 68.4 | 70.1 | 71.2 | 69.7 |

**Choice of distance**. Using cosine distance avoids introducing additional tuning parameter as the features are normalized before computing the distance. This and the usage of logits are just design choices. Other distance measures and intermediate features can be used together with our framework as well. We will explain this in the updated paper. As suggested by the reviewer, we will also introduce label leaking earlier in the introduction for clarity. We will release our trained model together with code as suggested.

[1] N. Narodytska and S. Kasiviswanathan. Simple black-box adversarial attacks on deep neural networks, CVPRW17


[Meta-Review · NeurIPS 2019]

After the rebuttal, all the reviewers agree that the paper is interesting and should be accepted. However, each of them have recommendations how to improve the paper, so the authors are encouraged to make these changes. This includes, among others, open sourcing the code and the models, correcting the results corresponding to black-box attacks, running experiments using additional gradient free attacks, corrections to the multiple restarts settings, and further ablation studies. Furthermore, during the discussions there was an agreement that Theorem 1 is meaningless and should be removed: since the concept "coupled" is not defined formally, one can pick any target and choose a penalty R=target-average loss. Also, any uncoupled regularizer can be made coupled by adding an arbitrarily small coupled term, which would not affect the behavior of the method. Therefore, the authors are encouraged to remove the current (almost) formal theorem statement and rather explain (perhaps as a theorem) how the coupling happens.